# Herbicide Screening and Application Method Development for Sustainable Weed Management in *Tagetes erecta* L. Fields

**DOI:** 10.3390/plants14111572

**Published:** 2025-05-22

**Authors:** Yiping Zhang, Dongyan Feng, Chengcheng Jia, Wangqi Huang, Feng Xu, Yalian Jiang, Junhong Huang, Ye Li, Jihua Wang, Dongsheng Tang

**Affiliations:** 1Flower Research Institute, Yunnan Academy of Agricultural Sciences, Kunming 650205, China; zyp@yaas.org.cn (Y.Z.); hwq@yaas.org.cn (W.H.); xf@yaas.org.cn (F.X.); jyl@yaas.org.cn (Y.J.); 2Yunnan Flower Breeding Key Laboratory, Yunnan Academy of Agricultural Sciences, Kunming 650205, China; 3Yunnan Flower Technology Innovation Center, Yunnan Academy of Agricultural Sciences, Kunming 650205, China; 4College of Plant Protection, Yunnan Agricultural University, Kunming 650201, China; a2024001@ynau.edu.cn (D.F.); 15725823903@163.com (C.J.); 5Yunnan Bohao Biotechnology Group Co., Ltd., Qujing 655331, China; 13887411022@163.com (J.H.); li_0618910@163.com (Y.L.)

**Keywords:** weed management, pre-emergence herbicides, post-emergence herbicides, crop safety

## Abstract

Marigold (*Tagetes erecta* L.), a crop of significant medicinal, ornamental, and economic value, faces severe industrialization challenges due to weed-induced yield losses (up to 60%). This study aims to identify safe and highly efficient herbicides for marigold, assess their effects on dominant weeds and crop safety, and provide a practical basis for large-scale cultivation. We evaluated 11 pre-emergence herbicides, 13 post-emergence herbicides, and agronomic practices (plastic mulch) through three field trials to optimize weed control, crop safety, and productivity. In Experiment 1, pre-emergence applications of pendimethalin (35% SC) and oxyfluorfen (240 g/L EC) under plastic mulch suppressed 85–99% of grass and broad-leaved weeds, elevating marigold yield to 1655.6 kg/667 m^2^ and increasing lutein content by 10.7% compared to controls, with no phytotoxicity to subsequent wheat (*Triticum aestivum* L.)or broad beans (*Vicia faba* L.). Experiment 2 demonstrated that post-cultivation soil treatment with metolachlor · oxyfluorfen · pendimethalin (50% EC) enhanced weed suppression (47.8–53.6%) and yield (3.4% increase) while ensuring crop safety. Experiment 3 revealed that the post-emergence herbicides haloxyfop-P-methyl (108 g/L EC) and fomesafen (250 g/L SL) achieved over 92% reduction in grass weed biomass and over 75% reduction in broadleaf weed density, respectively, alongside a 6.1% yield improvement. Therefore, region-specific strategies are recommended based on local agronomic conditions: high-value production zones should adopt integrated systems combining plastic mulch with pre-emergence herbicides; arid lands with extended crop rotation intervals require pre-emergence herbicides after intertillage and earthing-up; labor-abundant regions can rotate targeted post-emergence herbicides to delay resistance evolution. This study provides data-driven optimization strategies for comprehensive weed management in marigold fields, offering practical solutions to enhance industrial productivity and ecological sustainability.

## 1. Introduction

Marigold (*Tagetes erecta* L.), an annual herb in the Asteraceae family [1], has been cultivated globally as an economically valuable crop in China [2]. This species serves multiple industrial purposes, including landscaping, pharmaceutical, and cosmetic production, as well as functional food and feed additives [3,4]. Its extracts also show promise as natural biopesticides [3]. Native to Mexico, marigold was introduced to China in the 1990s and is now extensively cultivated in provinces such as Shandong, Jilin, Sichuan, and Yunnan due to its ecological resilience, broad adaptability, extended flowering duration, and diverse cultivars with vibrant floral characteristics.

However, weed infestation poses a severe threat to the industrialization of marigold cultivation, with yield losses reaching up to 60% under ineffective control [5]. Although no herbicides have been officially registered for weed management in marigold fields in China, traditional reliance on manual weeding or non-selective herbicides remains inefficient. These practices also carry risks of phytotoxicity and may accelerate the evolution of herbicide resistance. Notably, studies report that marigolds exhibit high sensitivity to broad-spectrum herbicides containing 2,4-dichlorophenoxyacetic acid (2,4-D). Key growth parameters—including dry weight, plant height, and flower count—were significantly reduced under sublethal doses of 2,4-D, with injury severity showing a positive correlation to herbicide concentration [6].

Pre-emergence herbicides are applied to the soil to suppress weed germination. Their advantages include long-lasting residual activity, broad-spectrum weed control, and minimal phytotoxicity to transplanted crops [7]. However, their efficacy is strongly influenced by factors such as soil moisture, organic matter content, and surface mulch conditions. For instance, a study demonstrated that the combination of dinoterb with oil palm frond (OPF) mulch exhibited the strongest synergistic effect, achieving 60–100% inhibition across all tested weed species. This outcome was significantly higher than when either dinoterb (0–50%) or OPF mulch (0–60%) was applied alone. In contrast, oxyfluorfen combined with OPF mulch only enhanced the inhibition of *Tridax procumbens* (70–90% vs. 20–40% with oxyfluorfen alone), while isoxaflutole combined with OPF showed no additional inhibitory effects except on *T. procumbens* [8]. Although mulch residues effectively reduce herbicide leaching, their efficacy is co-regulated by molecular properties (e.g., adsorption capacity and degradation rate) and rainfall intensity. For example, they exhibited greater leaching into deeper soil layers under intense rainfall, whereas glyphosate remained retained in the surface soil due to its strong adsorption capacity [9]. In contrast, post-emergence herbicides, which act through foliar absorption, offer rapid weed control and flexible dosage adjustments. However, long-term over-reliance on these herbicides significantly elevates resistance risks. In Argentina’s rain-fed cropping systems, the incidence of resistance to post-emergence herbicides is nine-fold higher than that to pre-emergence herbicides (90% vs. 10%). Resistance has expanded across multiple modes of action, including ALS inhibitors(Inhibition of acetolactate synthase) (29% of resistant species), auxin mimics (17%), and ACC’ase inhibitors (Inhibition of acetyl co-enzyme A carboxylase) (8%) [10]. Thus, herbicide selection and application require balancing multiple factors. Pre-emergence herbicides demand optimization of soil–herbicide–mulch interactions to balance between residual efficacy and ecological risks. In contrast, post-emergence strategies rely on precise application timing and proactive resistance management.

Therefore, this study aims to systematically evaluate the effects of herbicides (11 pre-emergence and 13 post-emergence) and agronomic practices (plastic mulch) on weed control efficacy, crop safety, and production outcomes in marigold fields through field trials. First, plastic mulch (Experiment 1) and post-cultivation soil treatments (Experiment 2) were compared to assess their differential impacts on weed suppression and crop safety. Second, selective herbicides were evaluated for their targeted control of grass and broadleaf weeds (Experiment 3). Finally, the impacts of different treatments on marigold yield and lutein content were quantified to propose region-specific weed management strategies that balance agronomic efficiency with ecological and economic sustainability.

## 2. Materials and Methods

### 2.1. Experimental Site and Design

The two experimental sites were located in Lingjiao Township (25.9963 N, 103.5998 E) and Deze Township (25.9010 N, 103.6883 E) in Qujing City, Yunnan Province, China (see Figure 1). These sites were designated as experimental site 1 and experimental site 2, respectively. The soil in this area is characterized as red soil with medium fertility. At experimental site 1, the planting system involved a wheat (*Triticum aestivum* L.)-marigold rotation, while experimental site 2 utilized a radish (*Raphanus sativus* L.)–marigold rotation. Notably, no herbicides were applied during the marigold planting process. Before the experiment commenced, during the marigold flowering period, we investigated the composition and structure of weed communities in the marigold fields using the quadrat method at both experimental sites. In each site, three fields (each larger than 667 m^2^) were selected, and three quadrats measuring 1 m × 1 m were established within each field (*n* = 18). The results indicated that the dominant weeds in the area were broad-leaved species, including *Galinsoga quadriradiata* Ruiz & Pavon, *Ageratum conyzoides* L., and *Acroglochin persicarioides* (Poir.) Moq, Additionally, the main grassy weed identified was *Digitaria sanguinalis* (L.) Scop. (see Table 1). The overview of marigold cultivation and the ourrence of weeds in the field were illustrated in Figure 2. At different growth stages of marigolds from May to August 2022, we conducted three separate experiments in both experimental sites. Experiment 1 involved a pre-emergent soil treatment using 11 herbicides, as shown in Table 2. At experimental site 1, the soil was irrigated using sprinkler belts before applying the black plastic film. Once the soil was sufficiently moistened, it underwent treatment procedures. Similarly, at site 2, the soil was irrigated with sprinkler belts prior to the herbicide treatment, but no mulch film was used. After these preparatory steps, marigold seedlings were transplanted. In Experiment 2, soil treatment was conducted after intertillage and earthing-up, approximately 45 days after the marigold transplanting in two experimental sites. The herbicides applied included all types tested in Experiment 1, with the addition of trifluralin. In Experiment 3, a post-emergent treatment was applied using 13 herbicides, as listed in Table 3, approximately 30 days after cultivation and earthing up in two experimental sites. The meteorological data recorded during the implementation of the three experimental trials can be found in the spreadsheet titled “Meteorological Data” within the Appendix A. In all these three trials, manual weeding was not conducted throughout the entire growth cycle of marigolds. The marigold variety utilized in this study was “Bohao No.1”, supplied by Yunnan Bohao Biotechnology Co., Ltd. (Kunming, China) The planting density for marigold was 4 plants/m^2^, with a plant spacing of 50 cm and a row spacing of 50 cm. The plot area measured 2 m^2^ (2 m × 1 m), and each herbicide treatment comprised four plots: three treatment groups and one control group. All plots were arranged randomly, with 0.5 m protective rows between them. All treatments were applied using electric knapsack sprayers (JvmJun, manufactured by America JinJun Group Limited, Jinhua, China) equipped with sector sprinklers operating at 0.15 MPa at the nozzle.

### 2.2. Investigation of Weed Control Efficacy and Damage Degree of Marigold

In Experiment 1, we measured the species, density, and above-ground fresh weight of weeds in each plot after 20 days of treatment. In Experiments 2 and 3, we counted the weed species and density after 15 days of treatment, and then measured these variables, along with fresh weight, after 30 days of treatment. Finally, we calculated the efficacy of the number of plants control and the efficacy of fresh weight control for each treatment as the following formula:$$\mathrm{Control}\;\mathrm{Effect}\;=\frac{\mathrm{CK}-\mathrm{PT}}{\mathrm{CK}}\times 100\%$$
where PT indicates the number or fresh weight of weeds in the treatment plot, while CK refers to the same metrics in the control plot.

In addition, in accordance with the guidelines for pesticide field efficacy trials [11], we have categorized the levels of herbicide damage as follows:

Grade 1: Crops grow normally without any signs of damage;

Grade 2: Minor herbicide damage to crops, affecting less than 10%;

Grade 3: Moderate herbicide damage to crops, which can later recover without affecting yield;

Grade 4: Crop damage caused by herbicides was severe and challenging to recover from, reducing yield;

Grade 5: Serious crop phytotoxicity leads to an inability to recover, resulting in significant yield reduction or complete loss.

In experimental site 1, once the marigolds began flowering, three were randomly selected from each plot. Every 10 days, the fully blooming flowers from these marigolds were collected, and their number and fresh weight were recorded. At the end of the flowering period, total flower count and fresh weight for each marigold were counted, and the yield (kg/667 m^2^) and yield increasing rate were calculated as following formula. In addition, during the peak flowering period of the marigolds, 10 flowers were collected from each plot for the determination of lutein content, conducted by Yunnan Bohao Biotechnology Co., Ltd. (Kunming, China) The yield of marigolds in experimental site 2 was not assessed because seed production was necessary.$$\mathrm{Yield}\;\mathrm{Increasing}\;\mathrm{Rate}\;=\frac{\mathrm{YT}-\mathrm{CK}}{\mathrm{YT}}\times 100\%$$
where YT and CK indicate the yield of marigolds in the treatment plot and the control plot, respectively.

### 2.3. Security Evaluation of Different Herbicides Following Crops

After harvesting marigolds in all treatment plots in experimental site 1, we planted half wheat and half broad beans (*Vicia faba* L.) in each plot on 26 October 2022. The sowing rates were set at 50 grains of wheat and 16 grains of broad beans per square meter. At that point, it had been 170 days since the end of Experiment 1, 115 days since the end of Experiment 2, and 86 days since the end of Experiment 3. Following the planting, we recorded the seedling emergence rates of both wheat and broad beans. Additionally, 40 days after planting, we measured the height of wheat (*n* = 12) and broad beans (*n* = 9), as well as their above-ground fresh weight (*n* = 3) in each plot. In experimental site 2, after harvesting the marigolds, the land was left fallow for specific reasons, so we did not plant wheat and broad beans subsequently.

### 2.4. Data Analysis

To compare the effects of various herbicide treatments on weed control efficacy, marigold yield, and the height and fresh weight of wheat and broad beans, we conducted a one-way ANOVA on different response factors using the Tukey test. Before the data analysis, the Shapiro–Wilk test confirmed that the data followed a normal distribution, and Levene’s test showed that the variances were equal. All data were analyzed using IBM SPSS Statistics 23.0 (Armonk, NY, USA).

## 3. Results

### 3.1. Weed Control Efficacy, Damage Assessment of Marigold, and the Safety of Subsequent Crops in the Experiment 1

In Experiment 1, conducted in experimental site 1 under film-covered conditions, the plant control efficacy and fresh weight control efficacy of each herbicide against gramineous and broad-leaved weeds were consistently above 90%, with the exception of oxyfluorfen’s control efficacy against broad-leaved weeds, which was approximately 85%. Furthermore, herbicides such as metolachlor, acetochlor, clomazone · acetochlor, clomazone, and flumioxazin demonstrated significantly reductions in both the density and fresh weight of gramineous weeds compared to butralin, oxyfluorfen, and metolachlor · clomazone. Additionally, acetochlor, flumioxazin, and metolachlor · oxyfluorfen · pendimethalin exhibited marked effectiveness in decreasing the density and fresh weight of broad-leaved weeds relative to butralin, oxyfluorfen, and metolachlor · clomazone.

In experimental site 2, where no film covering was applied, metolachlor · oxyfluorfen · pendimethalin achieved the highest control efficacy against grassy weeds, with plant density control and fresh weight control efficacy ranging from 55% to 65%, significantly surpassing those of other herbicides (control efficacy below 50%). The control efficacy of acetochlor and metolachlor · oxyfluorfen · pendimethalin against broad-leaved weeds was the highest (approximately 65%), markedly exceeding that of other herbicides (below 60%) (Table 4).

In two separate investigations conducted at two test sites, treatments involving pendimethalin, butralin, and oxyfluorfen demonstrated the least phytotoxicity to marigold plants, with a recorded level of 1. Metolachlor followed, exhibiting a phytotoxicity level of 2. However, in experimental site 2, the combination treatment of metolachlor, oxyfluorfen, and pendimethalin also resulted in a phytotoxicity level of 1. There was no difference in plant height and stem diameter between the pendimethalin and butralin treatments and the control group at experimental site 1. Although the plant height and stem diameter in the oxyfluorfen treatment were significantly lower than those in the control group during the first survey, no difference was observed between the oxyfluorfen treatment and the control group during the second survey. In contrast, with the exception of pendimethalin, butralin, and oxyfluorfen treatments, all other treatments resulted in significantly reduced plant height and stem diameter compared to the control group at experimental site 1.

In experimental site 2, treatments with pendimethalin, butralin, oxyfluorfen, and the combination of metolachlor, oxyfluorfen, and pendimethalin showed no difference from the control group in two surveys. However, during the second survey, the metolachlor treatment for marigold stem diameter also exhibited no difference compared to the control group (Table 5).

There were no differences in the number of flowers per plant among the treatments of pendimethalin, metolachlor, butralin, and oxyfluorfen when compared to the control group. The yields of marigolds in the pendimethalin and oxyfluorfen treatments were comparable to those in the control group, while the yields in the other eight treatments were significantly lower. Furthermore, the yield increasing rate was significantly higher for the pendimethalin and oxyfluorfen treatments, with the highest lutein content observed in the oxyfluorfen treatment (Table 6).

To examine how different herbicides affect the growth of subsequent crops, we planted wheat and broad beans after applying all treatments in experimental site 1. The results showed that the various treatments did not affect the seedling emergence rate, plant height, or fresh weight of either wheat or broad beans when compared to the control treatment (Table 7).

### 3.2. Weed Control Efficacy, Damage Assessment of Marigold, and the Safety of Subsequent Crops in the Experiment 2

The metolachlor · oxyfluorfen · pendimethalin treatment demonstrated the most significant reductions in both the density and fresh weight of grassy and broadleaf weeds across two experimental sites. The effects of metolachlor and acetochlor on grassy weeds were similar to those observed with the metolachlor · oxyfluorfen · pendimethalin at experimental site 1. Although metolachlor and acetochlor treatments effectively reduced the fresh weight of broadleaf weeds at this site, their impact on broad-leaved weeds densities was significantly less than that of metolachlor · oxyfluorfen · pendimethalin.

At experimental site 2, there was no difference in the plant density effectiveness or fresh weight control between acetochlor and the combination of metolachlor, oxyfluorfen, and pendimethalin against grassy weeds. Additionally, the efficacy of pendimethalin and metolachlor treatments against density of grassy weeds was comparable to that of metolachlor · oxyfluorfen · pendimethalin, although the fresh weight of reduction was significantly lower.

In addition, in experimental site 2, the control efficacy of pendimethalin, metolachlor and acetochlor against broad-leaved weeds was significantly lower than that of metolachlor · oxyfluorfen · pendimethalin. However, in the second investigation, there was no difference in the control effect of acetochlor against broad-leaved weeds compared to the combination treatment (Table 8).

In the two surveys, none of the treatments caused damage to the growth of marigolds (phytotoxicity level = 1) in two experimental sites. Additionally, there was no difference in the height and stem diameter of marigold between the various treatments and the control group in both two sites (Table 9).

All treatments did not significantly affect the number of flowers per marigold plant in experimental site 1. However, the yield and yield increase rate for marigolds treated with metolachlor · oxyfluorfen · pendimethalin were the highest, significantly exceeding those observed in the acetochlor, clomazone · acetochlor, and clomazone treatments. Additionally, the lutein content in marigolds was highest in the metolachlor · oxyfluorfen · pendimethalin treatment (Table 10).

The effects of various treatments on the seedling emergence rate, plant height, and fresh weight of subsequent crops, such as wheat and broad beans, showed no differences compared to the control group (Table 11).

### 3.3. Weed Control Efficacy, Damage Assessment of Marigold, and the Safety of Subsequent Crops in Experiment 3

Eight selective herbicides were evaluated for their effectiveness against grassy weeds, focusing on weed density and fresh weight. Except for the pinoxaden emulsion treatment, all other treatments achieved a weed density reduction rate of over 90% in experimental site 1, showing no differences among them. In experimental site 2, however, all treatments except sethoxydim and pinoxaden emulsion achieved a weed density reduction rate of more than 80%, with no differences among these treatments either. Additionally, the pinoxaden emulsion treatment against fresh weight of grassy weeds was significantly lower than that of the other treatments in two test sites.

Among the five selective herbicides tested for controlling broad-leaved weeds, the fomesafen treatment showed the highest reduction rates in both the density and fresh weight of these weeds across two experimental sites. The reductions in weed density and fresh weight were significantly greater with fomesafen compared to the treatments using bromoxynil octanoate and carfentrazone-ethyl at experimental site 1. However, at test site 2, there was no difference in the density reduction in Broad-leaved weeds between the carfentrazone-ethyl and fomesafen treatments. In terms of fresh weight control efficacy in experimental site 2, bromoxynil octanoate and carfentrazone-ethyl were significantly less effective against Broad-leaved weeds than fomesafen (Table 12).

In two surveys, two types of selective herbicides were found to have no negative impact on the growth of marigolds, with a phytotoxicity level of 1 in both experimental sites. Furthermore, there was no difference in the effects of the various treatments on plant height and stem diameter when compared to the control group (Table 13).

Eight herbicides used for controlling gramineous weeds did not significantly affect the number of flowers per plant and the yield of marigolds when compared to the control group. Although there was no difference in the yield increase rate of marigolds among the treatments, all treatments demonstrated a positive effect on marigold yield. However, the lutein content of marigold was highest in the clodinafop-butyl treatment (Table 14).

The five herbicides tested against Broad-leaved weeds did not significantly affect the number of flowers per marigold plant when compared to the control treatment. However, the fomesafen treatment produced the highest yield, exceeding that of both the control and carfentrazone ethyl treatments. The other four treatments, excluding carfentrazone ethyl, also showed positive effects on yield, though no differences were noted among them. Additionally, the lutein content in marigolds treated with bromoxynil octanoate was the highest (Table 14).

All experimental treatments showed no impact on the seedling emergence rate, plant height, or fresh weight of wheat and broad beans when compared to the corresponding control treatments. This was true for both herbicide treatments designed to control gramineous weeds and those aimed at broad-leaved weeds (Table 15).

## 4. Discussion

### 4.1. Application Effect of Pre-Emergence Herbicides in Marigold Fields

Both Experiment 1 and Experiment 2 applied herbicides before weed emergence; however, the effectiveness of weed control varied significantly between the two experiments. In Experiment 1, various treatments were applied to gramineous weeds, primarily focusing on *Digitaria sanguinalis* and broad-leaved weeds, which included *Galinsoga quadriradiata*, *Ageratum conyzoides*, and *Acroglochin persicarioide*. The reduction in density and fresh weight of these weeds ranged from 85% to 99% at experimental site 1. At experimental site 2, the combination of metolachlor, oxyfluorfen, and pendimethalin treatment demonstrated the highest control effectiveness to these weeds, achieving about 65% reduction in weed density and fresh weight. The control efficacy of the other treatments against gramineous weeds was below 50%, while their efficacy against broad-leaved weeds was less than 60%. In contrast, Experiment 2 indicated that the control effects of each treatment on these plants ranged from 23% to 53%. This discrepancy may be attributed to the presence of plastic film on the ridges in experimental site 1, which significantly enhanced weed control in Experiment 1 [12]. Reed et al. revealed that impermeable mulches extended the residual activity period of fomesafen, achieving a 60% reduction in weed density, but concurrently elevated herbicide carryover risks that may jeopardize rotational crop safety [13]. In contrast, biodegradable herbicide-loaded mulches developed by Eelager et al. exhibited dual advantages: targeted herbicide release through controlled-release technology alongside improvements in soil physicochemical properties, though strict regulation of threshold dosage is required to prevent phytotoxicity [14]. The drip irrigation-functional film system innovated by Zhang et al. [15] successfully reduced herbicide application by 45% without compromising maize yields, substantiating the feasibility of synergistic agronomic practices. Regarding herbicide selection, Hand et al. [16] emphasized the necessity to consider meteorological factors—under polyethylene mulch coverage, glyphosate and 2,4-D (2,4-dichlorophenoxyacetic acid) demonstrated low environmental risks post-rainfall, whereas dicamba exhibited persistent contamination potential even after precipitation events.

The herbicides used in the different treatments of Experiment 1 and Experiment 2 were largely identical. The primary difference between the two experiments lies in the timing of herbicide application, which leads to varying outcomes. In Experiment 1, pendimethalin and oxyfluorfen not only did not adversely affect the growth of marigolds, but they also significantly increased the yield. Additionally, among all treatments, marigolds treated with oxyfluorfen exhibited the highest lutein content, with a 10.7% increase compared to the control group in test site 1. The significant increase in lutein content observed in the oxyfluorfen-treated group may be associated with oxidative stress-mediated regulation of secondary metabolism. Similar studies in *Salvia officinalis* L. have demonstrated that oxyfluorfen induces ROS accumulation and activates antioxidant responses [17]. In Experiment 1, it was found that the most effective herbicides for controlling weeds in marigold fields were pendimethalin and oxyfluorfen. When mulch film was not used, the combination of metolachlor, oxyfluorfen, and pendimethalin provided the highest level of weed control without harming the growth of the marigolds. In Experiment 2, while all treatments did not harm the growth of marigolds, the metolachlor · oxyfluorfen · pendimethalin treatment exhibited the highest weed control efficacy (47.8%~53.6%), yield increase, and lutein content. Furthermore, this treatment had no negative impact on subsequent crops. Therefore, the most effective herbicide for controlling weeds in marigold fields was the combination of metolachlor, oxyfluorfen, and pendimethalin when mulch film was not applied. These herbicides act through three distinct mechanisms: inhibiting fatty acid synthesis (compromising membrane integrity) [18], disrupting chloroplast function (inducing photooxidation) [19], and suppressing cell division (impairing root growth) [20,21], respectively, collectively broadening weed control spectra. Their non-overlapping phytotoxic pathways in marigolds preclude synergistic crop damage. This result demonstrates the synergistic effects of herbicide combinations. Existing studies indicate that pre-emergence herbicide combinations, such as the herbicides flumioxazin + *S*-metolachlor, flumioxazin + imazethapyr and pyroxasulfone + sulfentrazone showed excellent control (97–99%) to *Amaranthus hybridus* L. [22]. Similarly, the combination of pendimethalin and terbuthylazine offers better control of certain weeds compared to pendimethalin used alone [23]. In paddy fields, the efficacy of the combination of bensulfuron-methyl and pretilachlor for weed control was better than that of oxyfluorfen [24]. While standalone pendimethalin and oxyfluorfen achieved 85–98% weed suppression under film mulching in Experiment 1, such single-target strategies risk accelerating resistance evolution, as evidenced by global PPO inhibitor (Inhibition of protoporphyrinogen oxidase) resistance reports [25].

### 4.2. Application Effect of Post-Emergence Herbicides in Marigold Fields

The experimental results demonstrated that haloxyfop-P-methyl achieved the highest fresh weight suppression rate (>92%) against gramineous weeds among all tested herbicides, with no observed phytotoxicity to marigolds (phytotoxicity level 1). Furthermore, fomesafen exhibited significantly higher reductions in both the density and fresh weight of broad-leaved weeds compared to bromoxynil octanoate and carfentrazone ethyl, while also producing the highest marigold yield (significantly surpassing the control and carfentrazone ethyl treatments). Critically, all herbicides showed no adverse effects on marigold growth parameters or non-target crops, with no significant differences in germination rate, plant height, or fresh weight compared to controls. These findings highlight their efficacy in weed control alongside crop safety and yield benefits.

This trial demonstrated that haloxyfop-P-methyl exhibits high efficacy against gramineous weeds (mainly including *Digitaria sanguinalis*) in marigold fields. However, the long-term exclusive use of this herbicide may elevate resistance risks in marigold systems, as evidenced by resistance evolution in Chinese cotton fields—87% of 65 *D. sanguinalis* populations developed low to moderate resistance [26]. Although marigold fields currently lack chemical herbicide exposure (minimizing resistance pressure), resistance mechanisms observed in cotton fields underscore the need for proactive management. To balance short-term control and long-term sustainability, we recommend rotating haloxyfop-P-methyl with alternative herbicides (e.g., ALS inhibitors) and implementing regional resistance monitoring programs. Although fomesafen is a selective herbicide for soybean fields, our trial showed that at the recommended rate (412.5 g a.i. ha^−1^), it effectively controlled major broadleaf weeds in marigold fields, including *Galinsoga quadriradiata*, *Ageratum conyzoides*, and *Acroglochin persicarioides*, without causing phytotoxicity to marigolds. However, resistance to fomesafen has been reported in *Amaranthus retroflexus* [27] and *Ipomoea* spp. [28] in northeastern China. Additionally, its prolonged use may pose persistent functional impacts on soil microbial communities [29,30].

### 4.3. Comprehensive Comparative Analysis of the Three Experiments Conducted in This Study

The three experiments exhibited significant yield disparities in the control treatments: Experiment 1 (1655.6 kg/667 m^2^) > Experiment 2 (1359.5 kg/667 m^2^) > Experiment 3 (861.3–922.2 kg/667 m^2^). Herbicide application combined with plastic mulching before marigold transplantation yielded the highest economic returns. The next best results came from cultivation and earthing up during the seedling stage with herbicide. The lowest yields were seen in unmulched and uncultivated plots that only received herbicide. Herbicide efficacy varied across the trials: oxyfluorfen (2.6% yield increase) outperformed pendimethalin (0.9%) in Experiment 1, while metolachlor · oxyfluorfen · pendimethalin (3.4%) in Experiment 2 and fomesafen (6.1%) in Experiment 3 showed progressive improvements. These yield enhancements correlated directly with reduced weed density and subsequent competitive relief for marigold growth [31].

Pre-emergence mulching (Experiment 1) achieved superior weed control (85.5–99.2%) and peak yield (1655.6 kg/667 m^2^), yet incurred microplastic pollution risks [32] and high costs (CNY1787.98/ha in Northern China) [33] from non-degradable films. Non-mulched pre-emergence herbicide mixtures (Experiment 2) provided moderate control (47.8–53.6%) through multi-target synergism, reducing both plastic usage and single-herbicide loads, but exhibited weed escape risks post treatment and required extended safety intervals (compared to Experiment 1). Post-emergence targeted spraying (Experiment 3) demonstrated adaptive precision (6.1% yield gain) but heightened ecological risks through herbicide resistance development and non-target species impacts.

Implementation strategies should be context-specific: (1) High-value production zones (e.g., lutein extraction bases) should prioritize mulching systems integrated with mandatory plastic film recovery protocols; (2) Arid regions with extended crop rotation intervals are better suited for non-mulched herbicide combinations, complemented by mechanical weed control through intertillage; (3) Areas with sufficient labor reserves may adopt post-emergence targeted herbicide applications as interim solutions, provided strict rotation of herbicides with distinct mechanisms of action is implemented to delay resistance development.

## 5. Conclusions

Based on the findings from three distinct herbicide application strategies in *Tagetes erecta* L. fields, this study highlights timing, herbicide formulation, and integrated approaches in optimizing weed control while ensuring crop safety and environmental sustainability. Experiments 1 and 2 demonstrated that pre-emergent soil treatments, particularly pendimethalin (35% SC) and oxyfluorfen (240 g/L EC) combined with plastic mulching, achieved superior weed suppression (85–99%) and enhanced marigold yield and lutein content. In contrast, the metolachlor · oxyfluorfen · pendimethalin (50% EC) formulation applied post-earthing up (Experiment 2) or before transplanting marigold (Experiment 1 in test site 1) provided moderate control (23–70%) but required no mulching, making it economically viable for large-scale operations. Experiment 3, which focused on post-emergent herbicides such as fomesafen (250 g/L L) and haloxyfop-P-methyl (108 g/L EC), delivered rapid, targeted control (>90% for broad-leaved weeds, >95% for grasses) with no phytotoxicity, proving ideal for emergency interventions.

Our research has some limitations regarding both spatial and temporal scope. However, the comparative analysis of the three experiments highlights several trade-offs: Experiment 1 prioritized efficacy but raised concerns about plastic waste; Experiment 2 balanced cost and practicality but had weed escape risks post-treatment; Experiment 3 offered flexibility but demanded precise timing. To address these issues, we recommend (1) adopting pendimethalin/oxyfluorfen with biodegradable mulching for high-intensity systems; (2) employing metolachlor·oxyfluorfen·pendimethalin in non-mulched, large-scale fields; and (3) reserving post-emergent herbicides for acute weed outbreaks. Future efforts should prioritize herbicide rotation to mitigate resistance risks and validate these strategies across diverse agroecological contexts to ensure scalability. Additionally, the weeding effects of key herbicides can be optimized at different dose levels through pot experiments based on the herbicide screening above.

## Figures and Tables

**Figure 1 plants-14-01572-f001:**
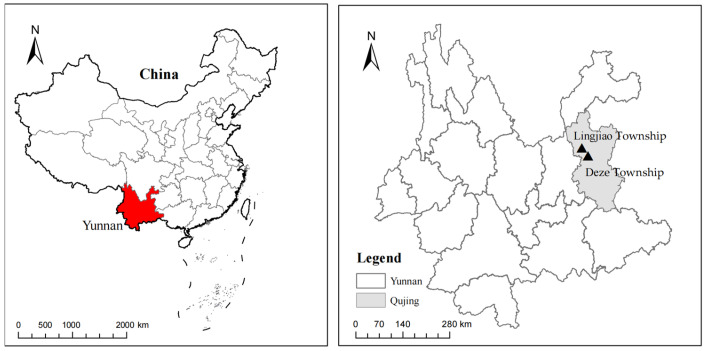
Location of the experimental sites.

**Figure 2 plants-14-01572-f002:**
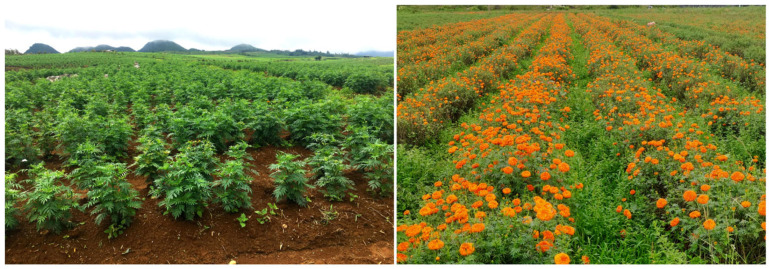
Overview of marigold (*Tagetes erecta* L.) cultivation and the occurrence of weeds in the field.

**Table 1 plants-14-01572-t001:** The weed density in the experimental fields (plants/m^2^).

Weeds	Experimental Site 1	Experimental Site 2	Average Value
*Digitaria sanguinalis*	12.44	44.22	28.33
*Setaria viridis*	0.33	0.33	0.33
*Eleusine indica*	0.56	1	0.78
*Cynodondactylon*	0.22	0.33	0.28
*Echinochloa crusgalli*	0.33	0.56	0.45
*Ageratum conyzoides*	0.56	16.22	8.39
*Galinsoga parviflora*	2.44	2.22	2.33
*Bidens pilosa*	1.67	0.67	1.17
*Sonchus oleraceus*	0.22	0.11	0.17
*Acalypha australis*	1	1.44	1.22
*Chenopodium album*	0.89	0.56	0.73
*Calystegia hederacea*	0.33	1.22	0.78
*Pharbitis purpurea*	1.11	0.22	0.67
*Capsella bursa-pastoris*	1	0.67	0.84
*Thlaspi arvense*	0	0.22	0.11
*Commelina communis*	0	1.78	0.89
*Vicia sepium*	0.56	0.56	0.56
*Pinellia ternata*	0	0.56	0.28
*Polygonum nepalense*	0	0.22	0.11
*Oxalis corniculata*	0	0.56	0.28
*Oenothera rosea*	0	0.22	0.11
*Acroglochin persicarioides*	40.89	0	20.45
*Siegesbeckia orientalis*	0.33	0	0.17

**Table 2 plants-14-01572-t002:** Characteristics of the herbicides used in the Experiment 1 and Experiment 2.

Herbicide Product	Active Ingredient	Mode of Action	Active Ingredient Rate (g a.i. ha^−1^)
Pendimethalin 35% SE	pendimethalin	Inhibition of microtubule assembly	787.5
Metolachlor 720 g/L EC	metolachlor	nhibition of very long chain fatty acid synthesis (VLCFA inhibitors)	1620
Acetochlor 900 g/L EC	acetochlor	VLCFA inhibitors	1080
Butralin 48% EC	butralin	Inhibition of microtubule assembly	1080
Oxyfluorfen 240 g/L EC	oxyfluorfen	Inhibition of protoporphyrinogen oxidase (PPO inhibitors)	162
Metolachlor · Clomazone 80% EC	64% metolachlor + 16% clomazone	Inhibition of deoxy-D-xyulose phosphate synthase (DOXP inhibitors) + VLCFA inhibitors	960
Clomazone · Acetochlor 500 g/L EC	acetochlor 400 g/L + clomazone 100 g/L	VLCFA inhibitors + DOXP inhibitors	450
Clomazone 360 g/L SE	clomazone	DOXP inhibitors	162
Flumioxazin 51% WE	flumioxazin	PPO inhibitors.	76.5
Metolachlor · Oxyfluorfen · Pendimethalin 50% EC	30% metolachlor + 5% oxyfluorfen + 15% pendimethalin	VLCFA inhibitors + PPO inhibitors + Inhibition of microtubule assembly	1125
Trifluralin 480 g/L EC (only for Experiment 2)	trifluralin	Inhibition of microtubule assembly	900
Control group	none	none	none

**Table 3 plants-14-01572-t003:** Characteristics of the herbicides used in the Experiment 3.

Herbicide Product		Active Ingredient	Mode of Action	Active Ingredient Rate (g a.i. ha^−1^)
Control for broad-leaved weeds	Bromoxynil octanoate 30% EC	bromoxynil octanoate	Inhibition of photosynthesis at photosystem II—D1 Histadine 215 binders (PS II Histadine 215 inhibitors)	373.5
	Carfentrazone-ethyl 10% WP	carfentrazone-ethyl	PPO inhibitors	34.5
	Bentazon 480 g/L SL	bentazone	PS II Histadine 215 inhibitors	1260
	Lactofen 240 g/L EC	lactofen	PPO inhibitors	82.5
	Fomesafen 250 g/L L	fomesafen	PPO inhibitors	412.5
Control for gramineae weeds	Sethoxydim 12.5% EC	sethoxydim	Inhibition of acetyl co-enzyme A carboxylase (ACC’ase inhibitors)	234.38
	Pinoxaden 5% EC	pinoxaden	ACC’ase inhibitors	52.5
	Fenoxaprop-P-ethyl 69 g/L EW	fenoxaprop-P-ethyl	ACC’ase inhibitors	67.28
	Clethodim 240 g/L EC	clethodim	ACC’ase inhibitors	115.2
	Clodinafop-butyl 22% OD	2% mesosulfuron-methyl + 20% clodinafop-propargyl	Inhibition of acetolactate synthase (ALS inhibitors) + ACC’ase inhibitors	67.5
	Quizalofop-P-ethyl 5% EC	quizalofop-P-ethyl	ACC’ase inhibitors	48.75
	Fluazifop-P-butyl 150 g/L EC	fluazifop-P-butyl	ACC’ase inhibitors	146.25
	Haloxyfop-P-methyl 108 g/L EC	haloxyfop-P-methyl	ACC’ase inhibitors	129.6
	Control group	none	none	none

**Table 4 plants-14-01572-t004:** Effects of different herbicides on weed control in Experiment 1.

Treatments	Gramineous Weeds		Broad-Leaved Weeds	
	Density Reduction Rate ± SD (%)	Fresh Weight Reduction Rate ± SD (%)	Density Reduction Rate ± SD (%)	Fresh Weight Reduction Rate ± SD (%)
Pendimethalin	95.4 ± 0.58 ab	97.84 ± 0.36 ^abc^	89.69 ± 1.32 ^d^	96.38 ± 0.54 ^ab^
Pendimethalin	43.57 ± 2.96 ^bcd^	43.36 ± 1.46 ^bc^	27.09 ± 1.44 ^e^	26.94 ± 2.38 ^g^
Metolachlor	97.13 ± 1.52 ^a^	98.40 ± 0.95 ^ab^	94.66 ± 0.76 ^bc^	92.92 ± 0.99 ^bc^
Metolachlor	36.22 ± 2.08 ^def^	37.01 ± 2.49 ^c^	44.68 ± 1.92 ^c^	53.02 ± 0.92 ^e^
Acetochlor	97.70 ± 1.15 ^a^	98.69 ± 0.68 ^ab^	99.24 ± 0.38 ^a^	98.69 ± 0.67 ^a^
Acetochlor	55.91 ± 1.98 ^a^	49.66 ± 2.70 ^b^	63.67 ± 1.83 ^a^	68.71 ± 1.03 ^a^
Butralin	89.66 ± 1.72 ^c^	95.17 ± 1.21 ^cd^	90.08 ± 0.76 ^d^	90.39 ± 2.00 ^c^
Butralin	30.45 ± 2.24 ^fg^	28.84 ± 1.98 ^d^	29.75 ± 2.16 ^e^	31.80 ± 2.19 ^f^
Oxyfluorfen	92.53 ± 1.15 ^bc^	95.13 ± 0.27 ^cd^	88.55 ± 1.15 ^d^	85.49 ± 2.87 ^d^
Oxyfluorfen	27.03 ± 2.74 ^g^	22.99 ± 3.29 ^d^	35.95 ± 1.79 ^d^	50.15 ± 1.22 ^e^
Metolachlor · Clomazone	91.95 ± 1.52 ^bc^	94.65 ± 1.63 ^d^	93.13 ± 1.75 ^c^	92.51 ± 1.3 ^bc^
Metolachlor · Clomazone	31.23 ± 2.05 ^fg^	37.63 ± 1.29 ^c^	46.33 ± 1.79 ^c^	59.09 ± 1.20 ^cd^
Clomazone · Acetochlor	97.13 ± 0.58 ^a^	98.23 ± 0.59 ^ab^	97.71 ± 0.66 ^a^	96.68 ± 1.06 ^ab^
Clomazone · Acetochlor	41.21 ± 1.39 ^cde^	37.90 ± 2.18 ^c^	46.96 ± 2.60 ^c^	67.21 ± 0.69 ^ab^
Clomazone	98.28 ± 1.00 ^a^	98.78 ± 0.63 ^ab^	93.51 ± 1.01 ^c^	96.13 ± 0.85 ^ab^
Clomazone	35.70 ± 1.84 ^ef^	44.86 ± 3.52 ^bc^	48.99 ± 1.54 ^c^	62.62 ± 0.76 ^bcd^
Flumioxazin	97.70 ± 0.58 ^a^	98.38 ± 0.29 ^ab^	98.85 ± 0.66 ^a^	98.29 ± 1.18 ^a^
Flumioxazin	50.92 ± 2.78 ^ab^	48.36 ± 2.07 ^b^	56.58 ± 1.70 ^b^	63.84 ± 2.31 ^bc^
Metolachlor · Oxyfluorfen · Pendimethalin	96.55 ± 1.00 ^a^	95.94 ± 1.41 ^bcd^	99.24 ± 0.38 ^a^	97.97 ± 1.12 ^a^
Metolachlor · Oxyfluorfen · Pendimethalin	54.86 ± 3.68 ^a^	64.72 ± 2.40 ^a^	61.77 ± 2.34 ^ab^	69.60 ± 1.68 ^a^
CK	0 (58.00)	0 (15.25)	0 (87.33)	0 (15.58)
CK	0 (127.00)	0 (129.76)	0 (263.33)	0 (1087.67)

Notes: In the same herbicide treatment, the first row represents the weed control effect in experiment site 1, and the second row represents the weed control effect in experiment site 2. Values present the means of three replicates, within a column values with different superscript letter are significantly different at 0.05 level in the same experimental site. Values in the brackets indicate the means of three replicates of weed density plants per 2 m^2^ and the fresh weight of aboveground plants per 2 m^2^ in non-treated control plots. All data were recorded 20 days after treatment.

**Table 5 plants-14-01572-t005:** Effects of different herbicides on growth of marigold in Experiment 1.

	First Survey			Second Survey		
Treatments	Height (cm)	Stem Diameter (mm)	Phytotoxicity Level	Height (cm)	Stem Diameter (mm)	Phytotoxicity Level
Pendimethalin	23.78 ± 0.44 ^ab^	6.22 ± 0.04 ^a^	1	51.33 ± 0.33 ^b^	11.57 ± 0.24 ^a^	1
Pendimethalin	24.29 ± 0.51 ^a^	4.47 ± 0.11 ^a^	1	50.41 ± 0.52 ^a^	10.51 ± 0.22 ^a^	1
Metolachlor	20.89 ± 0.43 ^c^	4.71 ± 0.14 ^d^	2	51.13 ± 0.29 ^b^	10.76 ± 0.09 ^b^	2
Metolachlor	22.19 ± 0.49 ^ab^	3.81 ± 0.04 ^c^	2	47.41 ± 0.29 ^b^	10.14 ± 0.10 ^ab^	1
Acetochlor	13.57 ± 0.26 ^e^	2.60 ± 0.03 ^g^	5	29.71 ± 0.39 ^g^	5.68 ± 0.10 ^g^	5
Acetochlor	15.33 ± 0.20 ^d^	2.49 ± 0.01 ^f^	5	39.01 ± 0.54 ^c^	8.33 ± 0.20 ^d^	5
Butralin	24.39 ± 0.30 ^a^	6.26 ± 0.01 ^a^	1	54.20 ± 0.35 ^a^	11.69 ± 0.29 ^a^	1
Butralin	24.21 ± 0.81 ^ab^	4.47 ± 0.06 ^a^	1	50.44 ± 0.32 ^a^	10.54 ± 0.14 ^a^	1
Oxyfluorfen	22.84 ± 0.27 ^b^	5.77 ± 0.05 ^b^	1	54.92 ± 0.47 ^a^	11.69 ± 0.16 ^a^	1
Oxyfluorfen	23.23 ± 1.48 ^ab^	4.38 ± 0.1 ^ab^	1	51.09 ± 0.89 ^a^	10.64 ± 0.10 ^a^	1
Metolachlor · Clomazone	20.20 ± 0.40 ^c^	4.31 ± 0.06 ^e^	4	43.09 ± 0.14 ^c^	8.81 ± 0.12 ^c^	4
Metolachlor · Clomazone	22.65 ± 0.75 ^ab^	3.91 ± 0.10 ^c^	4	47.84 ± 0.49 ^b^	9.73 ± 0.05 ^bc^	3
Clomazone · Acetochlor	16.90 ± 0.38 ^d^	3.33 ± 0.06 ^f^	4	40.67 ± 0.57 ^d^	7.61 ± 0.17 ^e^	4
Clomazone · Acetochlor	18.47 ± 0.57 ^c^	3.38 ± 0.08 ^d^	4	37.87 ± 0.77 ^c^	9.45 ± 0.23 ^c^	4
Clomazone	20.52 ± 0.25 ^c^	4.84 ± 0.02 ^d^	4	43.87 ± 0.35 ^c^	8.19 ± 0.15 ^d^	4
Clomazone	22.61 ± 0.46 ^ab^	4.23 ± 0.10 ^b^	4	47.53 ± 0.31 ^b^	10.59 ± 0.07 ^a^	4
Flumioxazin	14.08 ± 0.19 ^e^	3.55 ± 0.10 ^f^	5	33.93 ± 0.29 ^f^	6.50 ± 0.03 ^f^	5
Flumioxazin	14.80 ± 0.38 ^d^	2.91 ± 0.06 ^e^	5	39.39 ± 0.10 ^c^	9.53 ± 0.17 ^c^	4
Metolachlor · Oxyfluorfen · Pendimethalin	20.85 ± 0.60 ^c^	5.47 ± 0.16 ^c^	3	35.58 ± 0.42 ^e^	6.80 ± 0.02 ^f^	3
Metolachlor · Oxyfluorfen · Pendimethalin	23.40 ± 0.24 ^ab^	4.52 ± 0.10 ^a^	1	51.75 ± 0.57 ^a^	10.70 ± 0.20 ^a^	1
CK	23.9 ± 0.35 ^ab^	6.34 ± 0.02 ^a^	1	54.20 ± 0.60 ^a^	11.67 ± 0.22 ^a^	1
CK	23.73 ± 0.24 ^ab^	4.54 ± 0.03 ^a^	1	51.80 ± 0.60 ^a^	10.42 ± 0.13 ^a^	1

Notes: In the same herbicide treatment, the first row represents the data of marigold growth in test field 1, and the second row represents the data of marigold growth in test field 2. Values present the means of three replicates; within a column values with different superscript letter are significantly different at 0.05 level. The first survey and second survey refers to 15 days and 30 days, respectively, after treatment. The phytotoxicity levels, ranging from 1 to 5, represent the following: Level 1 indicates that crops grow normally without any signs of damage; Level 2 signifies minor herbicide damage to crops, affecting less than 10%; Level 3 denotes moderate herbicide damage, which may allow for recovery without impacting yield; Level 4 reflects severe crop damage caused by herbicides, making recovery challenging and resulting in reduced yield; and Level 5 indicates serious phytotoxicity, leading to an inability to recover and resulting in significant yield reduction or complete loss.

**Table 6 plants-14-01572-t006:** Effects of different herbicides on yield of marigold in Experiment 1.

Treatments	Flowers per Plant	Yield (kg/667 m^2^)	Yield Increasing Rate %	Lutein Conten (g/kg)
Pendimethalin	51.22 ± 1.31 ^a^	1670.76 ± 26.49 ^ab^	0.90 ± 0.36 ^a^	14.69
Metolachlor	50.11 ± 2.73 ^a^	1592.80 ± 16.55 ^bc^	−3.95 ± 0.54 ^b^	15.15
Acetochlor	33.11 ± 0.62 ^c^	999.91 ± 33.50 ^f^	−65.61 ± 1.62 ^g^	15.33
Butralin	51.33 ± 2.67 ^a^	1567.01 ± 30.92 ^c^	−5.69 ± 1.39 ^b^	15.49
Oxyfluorfen	52.89 ± 1.72 ^a^	1693.88 ± 22.55 ^a^	2.59 ± 0.71 ^a^	15.87
Metolachlor · Clomazone	39.44 ± 1.90 ^b^	1184.00 ± 18.17 ^e^	−39.85 ± 1.03 ^d^	15.28
Clomazone · Acetochlor	42.67 ± 1.76 ^b^	1399.22 ± 21.82 ^d^	−18.33 ± 0.54 ^c^	15.32
Clomazone	32.89 ± 0.78 ^c^	1020.07 ± 19.16 ^f^	−62.33 ± 1.48 ^f^	14.72
Flumioxazin	33.78 ± 1.06 ^c^	1108.70 ± 21.26 ^e^	−49.34 ± 0.85 ^e^	15.28
Metolachlor · Oxyfluorfen · Pendimethalin	44.22 ± 1.13 ^b^	1427.38 ± 26.45 ^d^	−16.00 ± 0.48 ^c^	15.04
CK	52.33 ± 2.03 ^a^	1655.64 ± 63.74 ^abc^	-	14.33

Notes: Values present the means of three replicates, within a column values with different superscript letter are significantly different at 0.05 level.

**Table 7 plants-14-01572-t007:** Effects of different herbicides on growth of following crops in Experiment 1.

	Wheat (*Triticum aestivum*)			Broad Bean (*Vicia faba*)		
Treatments	Seedling Emergence Rate (%)	Plant Heigh (cm)	Fresh Weight (g)	Germination Rate (%)	Plant Heigh (cm)	Fresh Weight (g)
Pendimethalin	79.33 ± 4.37 ^a^	15.05 ± 0.26 ^a^	2.09 ± 0.10 ^a^	60.42 ± 5.51 ^a^	8.33 ± 0.14 ^a^	7.28 ± 0.24 ^a^
Metolachlor	73.33 ± 6.96 ^a^	14.88 ± 0.25 ^a^	2.17 ± 0.03 ^a^	79.17 ± 7.51 ^a^	8.48 ± 0.12 ^a^	7.73 ± 0.31 ^a^
Acetochlor	74.67 ± 5.70 ^a^	14.98 ± 0.23 ^a^	2.13 ± 0.08 ^a^	79.17 ± 4.17 ^a^	8.45 ± 0.07 ^a^	7.42 ± 0.30 ^a^
Butralin	78.00 ± 8.33 ^a^	14.57 ± 0.13 ^a^	2.24 ± 0.18 ^a^	64.58 ± 11.60 ^a^	8.46 ± 0.12 ^a^	7.54 ± 0.24 ^a^
Oxyfluorfen	62.00 ± 5.29 ^a^	14.67 ± 0.14 ^a^	2.08 ± 0.20 ^a^	66.67 ± 5.51 ^a^	8.10 ± 0.28 ^a^	7.56 ± 0.43 ^a^
Metolachlor · Clomazone	67.33 ± 4.81 ^a^	14.78 ± 0.16 ^a^	2.21 ± 0.07 ^a^	77.08 ± 5.51 ^a^	8.35 ± 0.13 ^a^	6.92 ± 0.38 ^a^
Clomazone · Acetochlor	64.67 ± 5.21 ^a^	14.61 ± 0.15 ^a^	2.10 ± 0.12 ^a^	72.92 ± 2.08 ^a^	8.57 ± 0.09 ^a^	7.54 ± 0.09 ^a^
Clomazone	74.67 ± 5.30 ^a^	14.97 ± 0.18 ^a^	2.10 ± 0.06 ^a^	64.58 ± 11.02 ^a^	8.50 ± 0.14 ^a^	7.55 ± 0.35 ^a^
Flumioxazin	60.67 ± 2.40 ^a^	14.67 ± 0.22 ^a^	2.00 ± 0.10 ^a^	68.75 ± 3.61 ^a^	8.40 ± 0.17 ^a^	7.44 ± 0.17 ^a^
Metolachlor · Oxyfluorfen · Pendimethalin	76.67 ± 6.36 ^a^	14.95 ± 0.13 ^a^	2.06 ± 0.15 ^a^	64.58 ± 7.51 ^a^	8.10 ± 0.17 ^a^	7.50 ± 0.63 ^a^
CK	70.67 ± 3.71 ^a^	14.84 ± 0.17 ^a^	2.20 ± 0.06 ^a^	68.75 ± 9.55 ^a^	8.50 ± 0.14 ^a^	7.53 ± 0.53 ^a^

Notes: Values present the means of three replicates, within a column values with the same superscript letter are not significantly different at 0.05 level.

**Table 8 plants-14-01572-t008:** Effects of different herbicides on weed control in Experiment 2.

	Gramineous Weeds			Broad-Leaved Weeds		
Treatments	Density Reduction Rate ± SD (%) at First Survey	Density Reduction Rate ± SD (%) at Second Survey	Fresh Weight Reduction Rate ± SD (%) at Second Survey	Density Reduction Rate ± SD (%) at First Survey	Density Reduction Rate ± SD (%) at Second Survey	Fresh Weight Reduction Rate ± SD (%) at Second Survey
Pendimethalin	44.44 ± 4.2 ^abc^	44.78 ± 2.99 ^ab^	45.54 ± 2.54 ^bc^	28.57 ± 4.73 ^f^	29.31 ± 1.72 ^e^	35.19 ±2.32 ^e^
Pendimethalin	71.78 ± 2.92 ^abc^	69.03 ± 1.59 ^ab^	7.17 ± 2.17 ^bc^	64.47 ± 1.9 ^cd^	64.58 ± 2.37 ^bcd^	52.39 ± 4.85 ^c^
Metolachlor	47.62 ± 0.00 ^ab^	47.76 ± 1.49 ^a^	49.74 ± 0.38 ^ab^	46.43 ± 3.09 ^bc^	44.83 ± 3.45 ^bcd^	47.35 ± 1.00 ^abc^
Metolachlor	72.04 ± 1.62 ^abc^	64.43 ± 2.22 ^abc^	57.15 ± 2.45 ^bc^	67.07 ± 2.83 ^bcd^	64.02 ± 1.55 ^cd^	63.78 ± 3.61 ^b^
Acetochlor	44.44 ± 1.59 ^abc^	43.28 ± 2.99 ^ab^	47.32 ± 1.72 ^abc^	48.21 ± 4.73 ^ab^	46.55 ± 1.72 ^b^	49.30 ± 1.47 ^a^
Acetochlor	73.85 ± 0.69 ^ab^	68.3 ± 1.59 ^ab^	58.93 ± 2.30 ^ab^	72.85 ± 2.69 ^b^	70.64 ± 2.48 ^ab^	69.73 ± 0.92 ^ab^
Butralin	38.10 ± 2.75 ^bc^	38.81 ± 3.95 ^ab^	42.81 ± 1.69 ^cd^	23.21 ± 1.79 ^f^	24.14 ± 4.56 ^e^	28.51 ± 3.32 ^f^
Butralin	62.2 ± 2.47 ^de^	57.41 ± 1.98 ^cde^	50.83 ± 1.92 ^c^	34.53 ± 2.88 ^f^	34.09 ± 1.83 ^f^	40.01 ± 2.78 ^d^
Oxyfluorfen	39.68 ± 4.20 ^bc^	37.31 ± 2.59 ^ab^	44.07 ± 2.55 ^cd^	41.07 ± 0.00 ^cde^	41.38 ± 3.45 ^bcd^	40.58 ± 2.09 ^de^
Oxyfluorfen	55.73 ± 2.94 ^e^	54.26 ± 2.33 ^de^	51.78 ± 1.69 ^cd^	66.27 ± 1.77 ^bcd^	65.15 ± 1.06 ^bcd^	63.96 ± 2.92 ^b^
Metolachlor · Clomazone	41.27 ± 1.59 ^abc^	40.3 ± 2.99 ^ab^	44.7 ± 1.82 ^bcd^	42.86 ± 3.57 ^bcd^	43.1 ± 2.99 ^bcd^	43.45 ± 2.48 ^bcd^
Metolachlor · Clomazone	67.12 ± 1.44 ^bcd^	65.40 ± 1.75 ^ab^	55.17 ± 1.76 ^bcd^	70.86 ± 1.4 ^bc^	69.51 ± 1.89 ^bc^	66.64 ± 2.60 ^b^
Clomazone · Acetochlor	36.51 ± 2.75 ^bcd^	37.31 ± 4.48 ^ab^	43.02 ± 1.55 ^cd^	37.50 ± 4.73 ^e^	39.66 ± 1.72 ^cd^	39.37 ± 3.53 ^de^
Clomazone · Acetochlor	65.05 ± 1.35 ^cd^	62.01 ± 2.07 ^bcd^	50.63 ± 2.36 ^c^	66.27 ± 2.22 ^bcd^	66.1 ± 1.15 ^bcd^	67.33 ± 3.27 ^b^
Clomazone	34.92 ± 4.2 ^bcd^	35.82 ± 1.49 ^bc^	39.45 ± 1.69 ^d^	39.29 ± 1.79 ^de^	37.93 ± 5.17 ^d^	42.71 ± 4.44 ^bcd^
Clomazone	61.42 ± 3.05 ^de^	57.41 ± 2.79 ^cde^	41.54 ± 0.50 ^e^	56.89 ± 1.38 ^e^	55.30 ± 2.83 ^e^	66.50 ± 2.73 ^b^
Flumioxazin	39.68 ± 3.18 ^bc^	37.31 ± 2.59 ^ab^	39.87 ± 1.49 ^d^	44.64 ± 4.73 ^bcd^	44.83 ± 1.72 ^bcd^	41.97 ± 4.14 ^cd^
Flumioxazin	62.46 ± 1.81 ^de^	56.2 ± 4.40 ^de^	39.6 ± 3.55 ^ef^	63.27 ± 1.40 ^d^	61.93 ± 2.15 ^d^	65.59 ± 4.79 ^b^
Metolachlor · Oxyfluorfen · Pendimethalin	50.79 ± 1.59 ^a^	47.76 ± 3.95 ^a^	51.63 ± 1.17 ^a^	53.57 ± 1.79 ^a^	53.45 ± 5.17 ^a^	50.88 ± 3.48 ^a^
Metolachlor · Oxyfluorfen · Pendimethalin	76.44 ± 2.30 ^a^	70.48 ± 1.75 ^a^	63.46 ± 1.57 ^a^	82.04 ± 1.58 ^a^	76.14 ± 1.18 ^a^	73.55 ± 3.81 ^a^
Trifluralin	26.98 ± 6.35 ^bcd^	26.87 ± 2.99 ^c^	31.37 ± 1.14 ^e^	26.79 ± 3.57 ^f^	27.59 ± 2.99 ^e^	35.19 ± 2.98 ^e^
Trifluralin	55.21 ± 2.88 ^e^	50.15 ± 2.94 ^e^	36.23 ± 2.62 ^f^	38.52 ± 3.21 ^f^	37.12 ± 0.83 ^f^	50.58 ± 3.26 ^c^
CK	0 (21.00)	0 (22.33)	0 (31.76)	0 (18.66)	0 (19.33)	0 (35.90)
CK	0 (128.75)	0 (137.75)	0 (1102.95)	0 (167.00)	0 (176.00)	0 (459.67)

Notes: In the same herbicide treatment, the first row represents the weed control effect in experiment site 1, and the second row represents the weed control effect in experiment site 2. Values present the means of three replicates, within a column values with different superscript letter are significantly different at 0.05 level. Values in the brackets indicate the means of three replicates of weed density plants per 2 m^2^ and the fresh weight of above-ground plants per 2 m^2^ in non-treated control plots. The first survey and second survey refers to 15 days and 30 days, respectively, after treatment.

**Table 9 plants-14-01572-t009:** Effects of different herbicides on growth of marigold in Experiment 2.

	First Survey			Second Survey		
Treatments	Height (cm)	Stem Diameter (mm)	Phytotoxicity Level	Height (cm)	Stem Diameter (mm)	Phytotoxicity Level
Pendimethalin	72.08 ± 0.58 ^a^	12.57 ± 0.02 ^a^	1	85.00 ± 2.04 ^a^	14.90 ± 0.27 ^a^	1
Pendimethalin	76.92 ± 1.26 ^a^	11.98 ± 0.07 ^a^	1	80.11 ± 1.57 ^a^	65.78 ± 1.06 ^a^	1
Metolachlor	71.42 ± 1.12 ^a^	12.69 ± 0.17 ^a^	1	87.11 ± 1.66 ^a^	14.44 ± 0.15 ^a^	1
Metolachlor	74.50 ± 3.88 ^a^	11.93 ± 0.25 ^a^	1	79.39 ± 1.33 ^a^	64.06 ± 0.36 ^a^	1
Acetochlor	72.00 ± 0.25 ^a^	12.46 ± 0.10 ^a^	1	85.22 ± 1.83 ^a^	14.83 ± 0.45 ^a^	1
Acetochlor	72.00 ± 2.84 ^a^	12.35 ± 0.24 ^a^	1	79.00 ± 2.00 ^a^	63.89 ± 3.07 ^a^	1
Butralin	71.92 ± 0.58 ^a^	12.45 ± 0.25 ^a^	1	84.56 ± 2.54 ^a^	14.77 ± 0.22 ^a^	1
Butralin	75.42 ± 2.02 ^a^	12.74 ± 0.31 ^a^	1	79.56 ± 0.91 ^a^	64.11 ± 1.24 ^a^	1
Oxyfluorfen	72.42 ± 1.42 ^a^	12.50 ± 0.08 ^a^	1	87.33 ± 0.84 ^a^	14.70 ± 0.33 ^a^	1
Oxyfluorfen	74.08 ± 4.26 ^a^	11.93 ± 0.08 ^a^	1	80.50 ± 0.63 ^a^	65.50 ± 0.67 ^a^	1
Metolachlor · Clomazone	72.00 ± 0.00 ^a^	12.68 ± 0.17 ^a^	1	83.33 ± 1.58 ^a^	14.69 ± 0.50 ^a^	1
Metolachlor · Clomazone	75.58 ± 3.03 ^a^	12.43 ± 0.29 ^a^	1	79.39 ± 1.45 ^a^	65.61 ± 3.61 ^a^	1
Clomazone · Acetochlor	71.17 ± 0.51 ^a^	12.54 ± 0.32 ^a^	1	83.89 ± 1.90 ^a^	14.93 ± 0.53 ^a^	1
Clomazone · Acetochlor	70.42 ± 0.74 ^a^	12.25 ± 0.31 ^a^	1	78.78 ± 0.39 ^a^	67.94 ± 4.75 ^a^	1
Clomazone	71.17 ± 0.22 ^a^	12.60 ± 0.16 ^a^	1	84.44 ± 0.48 ^a^	14.99 ± 0.13 ^a^	1
Clomazone	75.75 ± 1.52 ^a^	12.11 ± 0.31 ^a^	1	78.17 ± 1.13 ^a^	65.06 ± 1.53 ^a^	1
Flumioxazin	72.33 ± 0.85 ^a^	12.88 ± 0.09 ^a^	1	83.33 ± 0.39 ^a^	14.92 ± 0.54 ^a^	1
Flumioxazin	75.17 ± 2.21 ^a^	11.84 ± 0.22 ^a^	1	79.44 ± 1.66 ^a^	68.39 ± 2.37 ^a^	1
Metolachlor · Oxyfluorfen · Pendimethalin	73.58 ± 1.47 ^a^	12.87 ± 0.02 ^a^	1	84.33 ± 1.33 ^a^	14.58 ± 0.22 ^a^	1
Metolachlor · Oxyfluorfen · Pendimethalin	76.67 ± 2.63 ^a^	12.54 ± 0.58 ^a^	1	81.44 ± 1.28 ^a^	67.06 ± 4.79 ^a^	1
Trifluralin	71.75 ± 1.04 ^a^	12.52 ± 0.16 ^a^	1	85.33 ± 1.71 ^a^	14.53 ± 0.30 ^a^	1
Trifluralin	73.58 ± 3.94 ^a^	12.28 ± 0.08 ^a^	1	77.61 ± 3.11 ^a^	65.67 ± 3.18 ^a^	1
CK	71.08 ± 0.58 ^a^	12.96 ± 0.06 ^a^	1	85.67 ± 3.34 ^a^	14.59 ± 0.12 ^a^	1
CK	74.08 ± 3.06 ^a^	12.34 ± 0.21 ^a^	1	79.89 ± 0.59 ^a^	67.94 ± 1.26 ^a^	1

Notes: In the same herbicide treatment, the first row represents the data of marigold growth in test field 1, and the second row represents the data of marigold growth in test field 2. Values present the means of three replicates; within a column values with the same superscript letter are not significantly different at 0.05 level. The first survey and second survey refers to 15 days and 30 days, respectively, after treatment. The phytotoxicity level 1 indicates that crops grow normally without any signs of damage.

**Table 10 plants-14-01572-t010:** Effects of different herbicides on yield of marigold in Experiment 2.

Treatments	Flowers per Plant	Yield (kg/667 m^2^)	Yield Increasing Rate %	Lutein Conten (g/kg)
Pendimethalin	33.56 ± 1.64 ^a^	1374.85 ± 27.47 ^ab^	1.04 ± 2.00 ^abc^	16.35
Metolachlor	32.44 ± 2.04 ^a^	1362.76 ± 16.50 ^ab^	0.21 ± 1.22 ^abc^	16.02
Acetochlor	32.22 ± 1.09 ^a^	1339.28 ± 20.44 ^bc^	−1.56 ± 1.57 ^bc^	16.14
Butralin	32.67 ± 3.67 ^a^	1375.50 ± 14.53 ^ab^	1.14 ± 1.04 ^abc^	17.44
Oxyfluorfen	32.00 ± 1.17 ^a^	1397.44 ± 13.11 ^ab^	2.70 ± 0.91 ^ab^	16.55
Metolachlor · Clomazone	33.44 ± 2.86 ^a^	1351.49 ± 10.90 ^ab^	−0.61 ± 0.82 ^abc^	16.2
Clomazone · Acetochlor	33.33 ± 2.59 ^a^	1328.66 ± 12.55 ^c^	−2.34 ± 0.97 ^c^	17.77
Clomazone	32.11 ± 1.35 ^a^	1328.34 ± 25.20 ^c^	−2.42 ± 1.96 ^c^	16.37
Flumioxazin	34.00 ± 1.02 ^a^	1373.90 ± 18.85 ^ab^	1.01 ± 1.35 ^abc^	17.1
Metolachlor · Oxyfluorfen · Pendimethalin	34.33 ± 1.76 ^a^	1407.79 ± 12.08 ^a^	3.42 ± 0.82 ^a^	17.8
Trifluralin	33.11 ± 1.06 ^a^	1343.84 ± 18.11 ^ab^	−1.20 ± 1.36 ^bc^	16.51
CK	32.67 ± 3.34 ^a^	1359.49 ± 30.10 ^ab^	-	17.14

Notes: Values present the means of three replicates, within a column values with different superscript letter are significantly different at 0.05 level.

**Table 11 plants-14-01572-t011:** Effects of different herbicides on growth of following crops in Experiment 2.

	Wheat (*Triticum aestivum*)			Broad Bean(*Vicia faba*)		
Treatments	Seedling Emergence Rate (%)	Plant Heigh (cm)	Fresh Weight (g)	Germination Rate (%)	Plant Heigh (cm)	Fresh Weight (g)
Pendimethalin	74.00 ± 16.17 ^a^	14.79 ± 0.15 ^a^	2.17 ± 0.19 ^a^	58.33 ± 9.08 ^a^	7.94 ± 0.15 ^a^	5.61 ± 0.80 ^a^
Metolachlor	76.67 ± 6.36 ^a^	14.73 ± 0.22 ^a^	2.06 ± 0.15 ^a^	62.50 ± 10.83 ^a^	7.96 ± 0.13 ^a^	5.38 ± 0.09 ^a^
Acetochlor	72.67 ± 13.28 ^a^	14.94 ± 0.17 ^a^	2.03 ± 0.09 ^a^	58.33 ± 9.08 ^a^	8.07 ± 0.15 ^a^	5.34 ± 0.23 ^a^
Butralin	73.33 ± 12.72 ^a^	15.10 ± 0.14 ^a^	2.13 ± 0.06 ^a^	62.50 ± 7.22 ^a^	8.08 ± 0.13 ^a^	5.37 ± 0.13 ^a^
Oxyfluorfen	68.00 ± 14.42 ^a^	14.89 ± 0.18 ^a^	2.10 ± 0.20 ^a^	58.33 ± 9.08 ^a^	8.07 ± 0.13 ^a^	5.35 ± 0.07 ^a^
Metolachlor · Clomazone	74.00 ± 10.58 ^a^	14.90 ± 0.18 ^a^	2.10 ± 0.15 ^a^	60.42 ± 9.08 ^a^	8.06 ± 0.17 ^a^	5.56 ± 0.34 ^a^
Clomazone · Acetochlor	68.67 ± 15.07 ^a^	14.75 ± 0.21 ^a^	2.17 ± 0.20 ^a^	58.33 ± 11.60 ^a^	8.10 ± 0.20 ^a^	5.47 ± 0.41 ^a^
Clomazone	68.67 ± 13.92 ^a^	14.87 ± 0.13 ^a^	2.05 ± 0.12 ^a^	56.25 ± 6.25 ^a^	8.01 ± 0.11 ^a^	5.32 ± 0.24 ^a^
Flumioxazin	76.00 ± 10.07 ^a^	14.87 ± 0.23 ^a^	2.09 ± 0.03 ^a^	60.42 ± 5.51 ^a^	8.14 ± 0.18 ^a^	5.75 ± 0.54 ^a^
Metolachlor · Oxyfluorfen · Pendimethalin	78.67 ± 8.67 ^a^	14.90 ± 0.21 ^a^	2.15 ± 0.03 ^a^	60.42 ± 15.02 ^a^	7.98 ± 0.14 ^a^	5.73 ± 0.16 ^a^
Trifluralin	77.33 ± 8.74 ^a^	14.70 ± 0.20 ^a^	2.05 ± 0.22 ^a^	58.33 ± 7.51	7.97 ± 0.14 ^a^	5.47 ± 0.48 ^a^
CK	79.33 ± 6.36 ^a^	15.02 ± 0.15 ^a^	2.14 ± 0.16 ^a^	60.42 ± 7.51 ^a^	8.30 ± 0.14 ^a^	5.61 ± 0.29 ^a^

Notes: Values present the means of three replicates, within a column values with the same superscript letter are not significantly different at the 0.05 level.

**Table 12 plants-14-01572-t012:** Effects of different herbicides on weed control in Experiment 3.

	Controlling for Gramineous Weeds		
Treatments	Density Reduction Rate ± SD (%) at First Survey	Density Reduction Rate ± SD (%) at Second Survey	Fresh Weight Reduction Rate ± SD (%) at Second Survey
Sethoxydim	94.38 ± 2.25 ^a^	93.58 ± 1.6 ^a^	92.05 ± 1.68 ^ab^
Sethoxydim	79.53 ± 3.09 ^ab^	79.66 ± 2.94 ^ab^	81.64 ± 1.88 ^b^
Pinoxaden emulsion	86.52 ± 2.92 ^b^	83.42 ± 3.25 ^b^	85.38 ± 0.89 ^c^
Pinoxaden emulsion	76.61 ± 1.17 ^b^	75.71 ± 2.26 ^b^	73.14 ± 1.6 ^c^
Fenoxaprop-P-ethyl	97.19 ± 1.12 ^a^	94.12 ± 0.54 ^a^	93.99 ± 0.42 ^a^
Fenoxaprop-P-ethyl	82.46 ± 2.68 ^ab^	83.62 ± 4.52 ^ab^	84.82 ± 1.48 ^ab^
Clethodim	95.51 ± 0.56 ^a^	94.65 ± 2.14 ^a^	93.95 ± 0.18 ^a^
Clethodim	83.63 ± 1.55 ^ab^	83.05 ± 1.70 ^ab^	85.07 ± 1.3 ^ab^
Clodinafop-butyl	96.07 ± 1.12 ^a^	93.05 ± 1.42 ^a^	92.8 ± 1.7 ^ab^
Clodinafop-butyl	83.04 ± 3.26 ^ab^	82.49 ± 2.46 ^ab^	86.24 ± 0.78 ^ab^
Quizalofop-P-ethyl	94.94 ± 1.69 ^a^	94.12 ± 1.07 ^a^	93.82 ± 1.91 ^a^
Quizalofop-P-ethyl	84.8 ± 0.59 ^a^	85.31 ± 0.57 ^a^	88.14 ± 0.34 ^a^
Fluazifop-P-butyl	92.7 ± 2.03 ^a^	90.91 ± 2.33 ^a^	89.84 ± 0.51 ^b^
Fluazifop-P-butyl	80.7 ± 2.68 ^ab^	81.36 ± 1.96 ^ab^	81.54 ± 2.09 ^b^
Haloxyfop-P-methyl	97.19 ± 0.56 ^a^	95.19 ± 0.93 ^a^	95.10 ± 0.54 ^a^
Haloxyfop-P-methyl	83.04 ± 1.55 ^ab^	83.62 ± 2.46 ^ab^	85.00 ± 1.04 ^ab^
CK	0 (59.33)	0 (62.33)	0 (75.46)
CK	0 (57.00)	0 (59.00)	0 (615.75)
	Controlling for Broad-leaved Weeds		
Bromoxynil octanoate	96.58 ± 0.9 ^b^	83.89 ± 1.33 ^b^	93.84 ± 1.64 ^bc^
Bromoxynil octanoate	73.23 ± 3.54 ^b^	74.23 ± 4.73 ^b^	76.45 ± 2.19 ^c^
Carfentrazone-ethyl	93.69 ± 0.79 ^c^	83.60 ± 2.34 ^b^	91.97 ± 0.36 ^c^
Carfentrazone-ethyl	78.28 ± 3.31 ^ab^	78.19 ± 0.5 ^ab^	82.89 ± 1.74 ^b^
Bentazon	97.3 ± 0.54 ^ab^	85.34 ± 0.81 ^ab^	92.61 ± 1.26 ^c^
Bentazon	83.84 ± 2.81 ^a^	79.18 ± 1.49 ^ab^	85.27 ± 0.88 ^ab^
Lactofen	98.02 ± 0.79 ^ab^	86.79 ± 1.05 ^ab^	96.54 ± 0.13 ^ab^
Lactofen	80.81 ± 1.82 ^ab^	79.68 ± 1.31 ^ab^	84.46 ± 1.11 ^ab^
Fomesafen	98.92 ± 0.31 ^a^	90.42 ± 1.01 ^a^	97.14 ± 0.33 ^a^
Fomesafen	83.33 ± 1.52 ^a^	83.15 ± 2.48 ^a^	88.96 ± 1.25 ^a^
CK	0 (185.00)	0 (229.66)	0 (194.33)
CK	0 (66.00)	0 (67.25)	0 (208.25)

Notes: In the same herbicide treatment, the first row represents the weed control effect in experiment site 1, and the second row represents the weed control effect in experiment site 2. Values present the means of three replicates, within a column values with different superscript letter are significantly different at 0.05 level in gramineous weeds or in Broad-leaved weeds. Values in the brackets indicate the means of three replicates of weed density plants per 2 m^2^ and the fresh weight of aboveground plants per 2 m^2^ in non-treated control plots. The first survey and second survey refers to 15 days and 30 days, respectively, after treatment.

**Table 13 plants-14-01572-t013:** Effects of different herbicides on growth of marigold in Experiment 3.

	First Survey on Treatments for Controlling Gramineous Weeds			Second Survey on Treatments for Controlling Gramineous Weeds		
Treatments	Height (cm)	Stem Diameter (mm)	Phytotoxicity Level	Height (cm)	Stem Diameter (mm)	Phytotoxicity Level
Sethoxydim	80.78 ± 0.87 ^a^	13.38 ± 0.08 ^a^	1	91.28 ± 1.84 ^a^	14.23 ± 0.25 ^a^	1
Sethoxydim	67.92 ± 0.96 ^a^	13.46 ± 0.15 ^a^	1	69.17 ± 0.17 ^a^	13.61 ± 0.14 ^a^	1
Pinoxaden emulsion	79.67 ± 2.52 ^a^	13.31 ± 0.47 ^a^	1	91.00 ± 1.39 ^a^	14.59 ± 0.20 ^a^	1
Pinoxaden emulsion	68.58 ± 0.33 ^a^	13.64 ± 0.02 ^a^	1	68.72 ± 1.66 ^a^	13.71 ± 0.09 ^a^	1
Fenoxaprop-P-ethyl	80.44 ± 1.47 ^a^	13.3 ± 0.29 ^a^	1	87.56 ± 2.02 ^a^	14.19 ± 0.33 ^a^	1
Fenoxaprop-P-ethyl	69.00 ± 0.29 ^a^	13.55 ± 0.04 ^a^	1	67.83 ± 1.83 ^a^	13.58 ± 0.13 ^a^	1
Clethodim	79.44 ± 3.28 ^a^	13.09 ± 0.39 ^a^	1	89.22 ± 1.66 ^a^	14.53 ± 0.26 ^a^	1
Clethodim	68.83 ± 1.04 ^a^	13.63 ± 0.38 ^a^	1	67.50 ± 0.10 ^a^	13.50 ± 0.06 ^a^	1
Clodinafop-butyl	82.56 ± 1.13 ^a^	13.22 ± 0.25 ^a^	1	89.67 ± 1.58 ^a^	14.60 ± 0.62 ^a^	1
Clodinafop-butyl	67.58 ± 1.16 ^a^	13.83 ± 0.11 ^a^	1	68.00 ± 1.15 ^a^	13.83 ± 0.06 ^a^	1
Quizalofop-P-ethyl	82.11 ± 0.97 ^a^	13.4 ± 0.44 ^a^	1	91.78 ± 2.96 ^a^	14.59 ± 0.74 ^a^	1
Quizalofop-P-ethyl	68.92 ± 0.55 ^a^	13.73 ± 0.20 ^a^	1	69.33 ± 1.53 ^a^	13.64 ± 0.14 ^a^	1
Fluazifop-P-butyl	80.89 ± 1.64 ^a^	13.39 ± 0.21 ^a^	1	88.33 ± 1.53 ^a^	14.23 ± 0.07 ^a^	1
Fluazifop-P-butyl	68.33 ± 0.44 ^a^	13.5 ± 0.05 ^a^	1	70.56 ± 2.47 ^a^	13.73 ± 0.16 ^a^	1
Haloxyfop-P-methyl	80.89 ± 2.30 ^a^	13.29 ± 0.27 ^a^	1	87.89 ± 2.23 ^a^	14.61 ± 0.39 ^a^	1
Haloxyfop-P-methyl	69.00 ± 0.80 ^a^	13.73 ± 0.21 ^a^	1	68.44 ± 0.87 ^a^	13.87 ± 0.13 ^a^	1
CK	81.33 ± 0.58 ^a^	13.32 ± 0.05 ^a^	1	87.67 ± 1.45 ^a^	14.63 ± 0.28 ^a^	1
CK	69.58 ± 1.04 ^a^	13.56 ± 0.05 ^a^	1	69.44 ± 0.56 ^a^	13.83 ± 0.07 ^a^	1
	First Survey on Treatments for Controlling Broad-leaved Weeds			Second Survey on Treatments for Controlling Broad-leaved Weeds		
Bromoxynil octanoate	80.00 ± 0.69 ^a^	13.56 ± 0.36 ^a^	1	91.67 ± 3.46 ^a^	14.88 ± 0.24 ^a^	1
Bromoxynil octanoate	67.83 ± 0.58 ^a^	13.56 ± 0.05 ^a^	1	69.11 ± 0.59 ^a^	13.47 ± 0.07 ^a^	1
Carfentrazone-ethyl	79.89 ± 0.87 ^a^	13.27 ± 0.46 ^a^	1	91.89 ± 3.76 ^a^	14.63 ± 0.02 ^a^	1
Carfentrazone-ethyl	65.61 ± 0.42 ^a^	13.68 ± 0.10 ^a^	1	69.00 ± 0.51 ^a^	13.72 ± 0.18 ^a^	1
Bentazon	80.33 ± 2.69 ^a^	13.49 ± 0.44 ^a^	1	91.89 ± 0.97 ^a^	14.61 ± 0.20 ^a^	1
Bentazon	67.35 ± 0.93 ^a^	13.60 ± 0.08 ^a^	1	70.28 ± 1.81 ^a^	13.73 ± 0.21 ^a^	1
Lactofen	80.11 ± 1.42 ^a^	13.64 ± 0.16 ^a^	1	90.33 ± 2.46 ^a^	14.59 ± 0.22 ^a^	1
Lactofen	66.53 ± 1.21 ^a^	13.51 ± 0.07 ^a^	1	69.22 ± 2.32 ^a^	13.66 ± 0.10 ^a^	1
Fomesafen	79.78 ± 1.85 ^a^	13.18 ± 0.23 ^a^	1	93.56 ± 0.48 ^a^	14.89 ± 0.25 ^a^	1
Fomesafen	67.42 ± 1.30 ^a^	13.58 ± 0.09 ^a^	1	70.78 ± 0.68 ^a^	13.72 ± 0.04 ^a^	1
CK	80.11 ± 0.73 ^a^	13.46 ± 0.33 ^a^	1	93.44 ± 2.12 ^a^	14.68 ± 0.20 ^a^	1
CK	66.33 ± 1.31 ^a^	13.69 ± 0.07 ^a^	1	70.11 ± 0.56 ^a^	13.76 ± 0.13 ^a^	1

Notes: In the same herbicide treatment, the first row represents the data of marigold growth in test field 1, and the second row represents the data of marigold growth in test field 2. Values present the means of three replicates; within a column values with the same superscript letter are not significantly different at 0.05 level in gramineous weeds or in Broad-leaved weeds. The first survey and second survey refers to 15 days and 30 days, respectively, after treatment. The phytotoxicity level 1 indicates that crops grow normally without any signs of damage.

**Table 14 plants-14-01572-t014:** Effects of different herbicides on yield of marigold in Experiment 3.

	Controlling for Gramineous Weeds			
Treatments	Flowers per Plant	Yield (kg/667 m^2^)	Yield Increasing Rate %	Lutein Conten (g/kg)
Sethoxydim	36.17 ± 1.20 ^a^	932.91 ± 9.93 ^a^	1.22 ± 2.54 ^a^	16.71
Pinoxaden emulsion	36.67 ± 0.88 ^a^	928.02 ± 6.23 ^a^	0.61 ± 0.67 ^a^	16.51
Fenoxaprop-P-ethyl	37.33 ± 1.45 ^a^	954.25 ± 14.33 ^a^	3.31 ± 1.45 ^a^	16.82
Clethodim	37.00 ± 1.53 ^a^	945.36 ± 8.66 ^a^	2.43 ± 0.89 ^a^	17
Clodinafop-butyl	36.50 ± 4.00 ^a^	928.91 ± 24.26 ^a^	0.58 ± 2.67 ^a^	18.57
Quizalofop-P-ethyl	38.67 ± 1.64 ^a^	972.93 ± 8.99 ^a^	5.19 ± 0.87 ^a^	17.4
Fluazifop-P-butyl	37.17 ± 1.88 ^a^	967.59 ± 21.18 ^a^	4.60 ± 2.09 ^a^	16.44
Haloxyfop-P-methyl	36.83 ± 0.44 ^a^	954.25 ± 25.22 ^a^	3.33 ± 1.12 ^a^	16.06
CK	36.67 ± 2.40 ^a^	922.24 ± 13.57 ^a^	-	16.3
	Controlling for Broad-leaved Weeds			
Bromoxynil octanoate	35.33 ± 1.17 ^a^	892.89 ± 25.48 ^ab^	3.37 ± 2.82 ^a^	19.11
Carfentrazone-ethyl	34.83 ± 1.88 ^a^	814.63 ± 18.55 ^c^	−5.84 ± 2.46 ^b^	19.09
Bentazon	35.17 ± 0.93 ^a^	871.55 ± 8.48 ^ab^	1.15 ± 0.95 ^a^	17.83
Lactofen	34.50 ± 1.53 ^a^	877.33 ± 12.33 ^ab^	1.79 ± 1.38 ^a^	17.41
Fomesafen	36.33 ± 0.88 ^a^	917.79 ± 10.07 ^a^	6.13 ± 1.02 ^a^	17.7
CK	34.67 ± 0.44 ^a^	861.32 ± 13.70 ^b^	-	18.16

Notes: Values present the means of three replicates, within a column values with different superscript letter are significantly different at 0.05 level.

**Table 15 plants-14-01572-t015:** Effects of different herbicides on growth of following crops in Experiment 3.

	Wheat (*Triticum aestivum*)			Broad Bean (*Vicia faba*)		
Treatments	Seedling Emergence Rate (%)	Plant Heigh (cm)	Fresh Weight (g)	Germination Rate (%)	Plant Heigh (cm)	Fresh Weight (g)
Sethoxydim	58.67 ± 7.86 ^a^	14.97 ± 0.17 ^a^	2.07 ± 0.18 ^a^	58.33 ± 2.08 ^a^	8.31 ± 0.22 ^a^	5.91 ±0.99 ^a^
Pinoxaden emulsion	64.67 ± 9.68 ^a^	14.91 ± 0.34 ^a^	2.15 ± 0.88 ^a^	60.42 ± 7.51 ^a^	8.25 ± 0.20 ^a^	5.94 ± 0.77 ^a^
Fenoxaprop-p-ethyl	70.67 ± 11.62 ^a^	15.16 ± 0.37 ^a^	2.27 ± 0.27 ^a^	56.25 ± 0.00 ^a^	8.37 ± 0.25 ^a^	6.06 ± 0.25 ^a^
Clethodim	57.33 ± 4.67 ^a^	15.03 ± 0.37 ^a^	2.31 ± 0.17 ^a^	62.50 ± 7.22 ^a^	8.16 ± 0.18 ^a^	5.86 ± 0.46 ^a^
Clodinafop-butyl	68.67 ± 9.40 ^a^	15.12 ± 0.41 ^a^	2.24 ± 0.74 ^a^	56.25 ± 7.22 ^a^	8.18 ± 0.20 ^a^	5.94 ± 0.57 ^a^
Quizalofop-p-ethyl	66.00 ± 5.77 ^a^	14.97 ± 0.23 ^a^	2.2. ± 0.15 ^a^	64.58 ± 5.51 ^a^	8.27 ± 0.23 ^a^	5.91 ± 0.32 ^a^
Fluazifop-p-butyl	64.67 ± 4.06 ^a^	14.72 ± 0.15 ^a^	2.15 ± 0.13 ^a^	60.42 ± 5.51 ^a^	8.44 ± 0.30 ^a^	5.96 ± 0.68 ^a^
Haloxyfop-p-methyl	64.67 ± 13.97 ^a^	15.08 ± 0.30 ^a^	2.15 ± 0.52 ^a^	64.58 ± 12.67 ^a^	8.25 ± 0.17 ^a^	5.80 ± 0.32 ^a^
CK	65.33 ± 2.40 ^a^	15.03 ± 0.32 ^a^	2.17 ± 0.57 ^a^	60.42 ± 5.51 ^a^	8.25 ± 0.19 ^a^	6.13 ± 0.17 ^a^
Bromoxynil octanoate	71.33 ± 8.74 ^a^	15.03 ± 0.21 ^a^	2.33 ± 0.23 ^a^	52.08 ± 8.33 ^a^	8.31 ± 0.15 ^a^	7.13 ± 0.33 ^a^
carfentrazone-ethyl	66.67 ± 9.68 ^a^	15.02 ± 0.20 ^a^	2.25 ± 0.11 ^a^	50.00 ± 6.25 ^a^	8.32 ± 0.17 ^a^	6.95 ± 0.48 ^a^
Bentazon	64.67 ± 15.76 ^a^	14.93 ± 0.14 ^a^	2.11 ± 0.10 ^a^	52.08 ± 5.51 ^a^	8.31 ± 0.18 ^a^	7.00 ± 0.54 ^a^
Lactofen	58.00 ± 10.26 ^a^	14.80 ± 0.20 ^a^	2.23 ± 0.05 ^a^	56.25 ± 7.22 ^a^	8.44 ± 0.26 ^a^	7.05 ± 0.29 ^a^
Fomesafen	67.33 ± 5.21 ^a^	14.81 ± 0.20 ^a^	2.11 ± 0.03 ^a^	56.25 ± 6.25 ^a^	8.42 ± 0.20 ^a^	6.97 ± 0.60 ^a^
CK	65.33 ± 7.06 ^a^	15.02 ± 0.23 ^a^	2.29 ± 0.17 ^a^	56.25 ± 12.50 ^a^	8.30 ± 0.15 ^a^	7.12 ± 0.66 ^a^

Notes: Values present the means of three replicates, within a column values with the same superscript letter are not significantly different at 0.05 level.

## Data Availability

The original data presented in the study are openly available in Dryad at DOI: 10.5061/dryad.fn2z34v66, accessed on 18 May 2025.

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
