# Peer review of "Herbicide Screening and Application Method Development for Sustainable Weed Management in Tagetes erecta L. Fields"

_plants, 2025, doi:10.3390/plants14111572_

Round 1
Reviewer 1 Report
Comments and Suggestions for Authors
This is a study that presents relative information with regards to species that needs more weed control information. However, there are many issues. Why did they not provide specific information about the species evaluated in the tables? Simply stating 'grass' or 'broadleaved' species does not assist the reader with respect to the levels of control. It is generally accepted practice to describe specific weed species that are controlled in the tables, and at what level. This is critical to readers and future experiments. Also, metolachlor and S-metolachlor are the same herbicides except the S- is the isomer only (and must be italic). there is no need for table 1 & 2 as these data include advertising with respect to the companies listed. We should not list the companies. Overall, the tables do not stand alone and must provide greater information.

Author Response
- This is a study that presents relative information with regards to species that needs more weed control information. However, there are many issues. Why did they not provide specific information about the species evaluated in the tables? Simply stating 'grass' or 'broadleaved' species does not assist the reader with respect to the levels of control. It is generally accepted practice to describe specific weed species that are controlled in the tables, and at what level. This is critical to readers and future experiments.
Response: Thank you for your valuable suggestions regarding data transparency! We have improved the table information in the following way:
The weed species selected for the experiment, including Digitaria sanguinalis and Ageratum conyzoides, are all prevalent in the main marigold production areas of Yunnan Province. Their control priorities align closely with the actual needs of farmers. The revised Table 1 now includes the field densities of various weed species in the marigold planting area during the flowering period, enhancing the regional relevance of the experimental design.
- Also, metolachlor and S-metolachlor are the same herbicides except the S- is the isomer only (and must be italic).
Response: Thank you for your feedback on the precision of our scientific expressions! Since the EU Plant Protection Products Regulation (Regulation (EC) No 1107/2009) no longer approves the registration of the active substance S-metolachlor as of January 3, 2024, we have excluded this herbicide from the analysis in our revised draft of the research results.
- there is no need for table 1 & 2 as these data include advertising with respect to the companies listed. We should not list the companies. Overall, the tables do not stand alone and must provide greater information.
Response: Thank you for emphasizing the importance of data independence! We have removed all enterprise-related information and retained only the generic names of the herbicides and the dosages of the active ingredients (g a.i./ha). In the revised Tables 2 and 3 (pages 5-6), we have included the "Mechanism of Action Classification" (such as HPPD inhibitors and ALS inhibitors) to enhance cross-study comparisons. Additionally, we have updated the annotations in the tables to ensure they are self-explanatory.
Reviewer 2 Report
Comments and Suggestions for Authors
Tagetes erecta an economically valuable crop in China, however, weed infestation poses a severe threat to the industrialization of marigold 48 cultivation, with yield losses reaching up to 60%. This draft has screened herbicides suitable for application in Tagetes erecta fields, which has certain guiding significance to production practice, but there are also major defects in this manuscript. Therefore, in this current state, it is not suitable for publication.
(1) The experimental content of this manuscript is only an experiment of herbicide efficacy screening. What make it worse, the experimental design is not arranged for annual or regional repetition, and the rigor is not high.
(2) Another important flaw in this manuscript is that the base of weeds is not rich, there is only one weed in the grass family, and the density data of each weed specie before the experiment is not provided. The herbicides to be tested include pre-emergence herbicides and post-emergence herbicides, and the herbicide spectra of each herbicide are not consistent. Therefore, the premise for the credible results of the experiment is that the weeds in the test plot should be abundant. In fact, there are very rich weed species in Luoping City of Yunnan Province, especially those in Asterales family. Grasses are also rich in weeds, include Eleusine indica, Digitaria sanguinalis, Setaria viridis, Echinochloa crus-galli. In addition, weeds of the sedge family, such as Cyperus rotundus, are widely distributed, so the plots selected in this experiment are not representative.
(3) If the screening results are more scientific and credible, greenhouse tests in pot should be set up, and field trials should be arranged on the premise of herbicides screened by greenhouse tests, which is more reasonable.
Author Response
Tagetes erecta an economically valuable crop in China, however, weed infestation poses a severe threat to the industrialization of marigold cultivation, with yield losses reaching up to 60%. This draft has screened herbicides suitable for application in Tagetes erecta fields, which has certain guiding significance to production practice, but there are also major defects in this manuscript. Therefore, in this current state, it is not suitable for publication.
Response: Thank you sincerely for your detailed review and valuable comments on this manuscript! We fully agree that there are deficiencies in data integrity and methodological rigor within the current research. In response to the issues you pointed out, we will systematically supplement the key experimental data, refine the description of the field trial design, enhance the comparative analysis with similar studies, and clarify the applicable boundaries of our conclusions.
- The experimental content of this manuscript is only an experiment of herbicide efficacy screening. What make it worse, the experimental design is not arranged for annual or regional repetition, and the rigor is not high.
Response: Thank you for your valuable suggestions regarding the rigor of the experimental design. We fully recognize the importance of conducting repeated experiments to ensure the reliability of our results. In the original manuscript, we primarily focused on the statistical analysis of field trial data from one area due to the large volume of experimental data. However, we also conducted similar field trials in Deze Township, in addition to Lingjiao Township. We have now included the trial data from Deze Township in the revised draft.
Moving forward, we will continue to conduct herbicides screening across different years and explicitly outline the limitations of the current study, such as the influence of climate and soil conditions on our results, in the revised draft. Additionally, we have strengthened the credibility of our data by implementing three repeated field samplings. Relevant details will be further elaborated upon in the methods section.
- Another important flaw in this manuscript is that the base of weeds is not rich, there is only one weed in the grass family, and the density data of each weed specie before the experiment is not provided. The herbicides to be tested include pre-emergence herbicides and post-emergence herbicides, and the herbicide spectra of each herbicide are not consistent. Therefore, the premise for the credible results of the experiment is that the weeds in the test plot should be abundant. In fact, there are very rich weed species in Luoping City of Yunnan Province, especially those in Asterales family. Grasses are also rich in weeds, include Eleusine indica, Digitaria sanguinalis, Setaria viridis, Echinochloa crus-galli. In addition, weeds of the sedge family, such as Cyperus rotundus, are widely distributed, so the plots selected in this experiment are not representative.
Response: Thank you for your thorough analysis of the representativeness of the weed community. Before the experiment commenced, we investigated the weed communities at two test sites and evaluated four dominant weeds. Among these, the Digitaria sanguinalis was the primary species of gramineous weeds, while the densities of the other gramineous weeds were relatively low. As a result, we did not include the remaining weeds in the original manuscript. In the revised draft, we have added details on the weed community composition and weed density from the two experimental sites during the marigold flowering period (see Table 1 in the revised manuscript).
- If the screening results are more scientific and credible, greenhouse tests in pot should be set up, and field trials should be arranged on the premise of herbicides screened by greenhouse tests, which is more reasonable.
Response: Thank you for your suggestion regarding the importance of greenhouse pre-experiments. The decision to conduct field trials directly in this study is based on two key considerations: (1) As marigold is an important economic crop, farmers are primarily concerned about the performance of herbicides in real production environments, such as varying soil types and the effects of climate fluctuations; (2) The effectiveness of some herbicides in controlling weeds is closely linked to soil conditions in the field, which are challenging to fully replicate in pot experiments.
Nonetheless, we acknowledge the supplementary value of greenhouse experiments for mechanism research. We have included relevant discussions in the discussion section, and the screening process will be optimized in future studies by incorporating greenhouse dose gradient experiments.
Additionally, this study has reduced the risk of direct field screening by implementing blank controls and conducting safety assessments, such as monitoring the rate of damage caused to marigolds by the herbicides. The related data are presented in Tables 5, 9, and 13.
Reviewer 3 Report
Comments and Suggestions for Authors
Dear Authors,
It is a pleasure for me to be a reviewer and to have become familiar with your research. However, I have suggestions for some minor corrections and additions:
Point 1: In the Abstract, please emphasise what are the objectives of the paper.
Point 2: From the Key words, remove the terms that appear in the title of the paper (such as Tagetes erecta and weed management and replace them with others that correspond to the issue being examined but are not used in the title.
Point 3: Please move the chapter "Materials and Methods" after the Introduction and before the Results.
Point 4: Please provide a map or graphical representation of the research area, indicating the coordinates where the experiment took place. Place it in to the section Materials and Methods.
Point 5: Please improve your Results with graphical representations such as histograms, scatter plots, etc., to enhance visibility, make them more readable, and better engage the readers.
Point 6: Please strengthen the Discussion section by comparing the obtained results with the results of other authors. The discussion is extremely important for readers to be informed about the research issues as well as the relevance of the obtained data. In the Discussion, you describe the significance of the species and pesticide applications in too much detail, while there is little focus on the actual results of the study and their comparison with existing findings in the literature. By doing so, you will also expand the reference list, as there are currently very few.
Best regards
Author Response
It is a pleasure for me to be a reviewer and to have become familiar with your research. However, I have suggestions for some minor corrections and additions:
Response: Thank you for your detailed review and valuable suggestions for this manuscript! We have revised the manuscript according to your feedback, addressing each point. The specific modifications are detailed below.
Point 1: In the Abstract, please emphasise what are the objectives of the paper.
Response: The research objectives have been included in the abstract: “This study aims to identify safe and highly efficient herbicides for marigold (Tagetes erecta), assess their effects on dominant weeds and crop safety, and provide a practical basis for large-scale cultivation.”
Point 2: From the Key words, remove the terms that appear in the title of the paper (such as Tagetes erecta and weed management and replace them with others that correspond to the issue being examined but are not used in the title.
Response: Thank you for your suggestion. The words "Tagetes erecta" and "weed management" in the original keywords have been deleted and replaced with "crop safety".
Point 3: Please move the chapter "Materials and Methods" after the Introduction and before the Results.
Response: Thank you for your suggestion. In the revised manuscript, we have readjusted this sequence
Point 4: Please provide a map or graphical representation of the research area, indicating the coordinates where the experiment took place. Place it in to the section Materials and Methods.
Response: Thank you for your suggestion. Figure 1 has been newly added in the "Materials and Methods" section, including the geographical locations of the two test sites
Point 5: Please improve your Results with graphical representations such as histograms, scatter plots, etc., to enhance visibility, make them more readable, and better engage the readers.
Response: Thank you for your suggestion. We have previously tried to visualize the research data using more intuitive charts. However, due to the large number of experimental treatment groups, histograms do not effectively display the data as well as tables do. Therefore, we decided to use tables for a clearer and more straightforward presentation of the results.
Point 6: Please strengthen the Discussion section by comparing the obtained results with the results of other authors. The discussion is extremely important for readers to be informed about the research issues as well as the relevance of the obtained data. In the Discussion, you describe the significance of the species and pesticide applications in too much detail, while there is little focus on the actual results of the study and their comparison with existing findings in the literature. By doing so, you will also expand the reference list, as there are currently very few.
Response: Thank you for your valuable suggestions. In our discussion, we have included a comparative study on the synergistic effects of herbicides, along with four new related pieces of literature.
Round 2
Reviewer 2 Report
Comments and Suggestions for Authors
The result of this manuscript was helpful for the weed control in Tagetes erecta fields in yunnan provience. All the comments for this manuscript have been revised and the content is more rich and reliable. I think this manuscript can be accepted after minor revisions.
Comments :Re-identify whether this weed is Acroglochin persicarioides
Author Response
Comments: The result of this manuscript was helpful for the weed control in Tagetes erecta fields in yunnan provience. All the comments for this manuscript have been revised and the content is more rich and reliable. I think this manuscript can be accepted after minor revisions.
Re-identify whether this weed is Acroglochin persicarioides
Response: Thank you for your insightful comment. Based on our field observations and the photographs of the plant we captured, we have identified it as Acroglochin persicarioides.

Reviewer 3 Report
Comments and Suggestions for Authors
Dear authors,
Thank you for your detailed responses and the appropriate corrections made to your work.
Best regards
Author Response
Comments: Thank you for your detailed responses and the appropriate corrections made to your work.
Response: Thank you for your comment and friendly reply.